# Identifying Irregular Financial Operations Using Accountant Comments and Natural Language Processing Techniques

**Vytautas Rudžionis [1,2,\*], Audrius Lopata [1], Saulius Gudas [1], Rimantas Butleris [1], Ilona Veitaitė [1], Darius Dilijonas [3], Evaldas Grišius [3], Maarten Zwitserloot [4] and Kristina Rudzioniene [2]**

1 Faculty of Informatics, Kaunas University of Technology, Studentu 50, LT-51368 Kaunas, Lithuania
2 Kaunas Faculty, Vilnius University, Muitines 8, LT-44280 Kaunas, Lithuania
3 Intellerts, UAB, Studentu g. 3A-9, LT-50232 Kaunas, Lithuania
4 Intellerts B.V., Europalaan 400-7, 3526 KS Utrectht, The Netherlands
\* Correspondence: vytautas.rudzionis@knf.vu.lt

**Featured Application: The paper presents application of natural language processing techniques on accountant left comments to identify potentially irregular financial operations.**

**Abstract:** Finding not typical financial operations is a complicated task. The difficulties arise not only due to the sophisticated actions of fraudsters but also because of the large number of financial operations performed by business companies. This is especially true for large companies. It is highly desirable to have a tool to reduce the number of potentially irregular operations significantly. This paper presents an implementation of NLP-based algorithms to identify irregular financial operations using comments left by accountants. The comments are freely written and usually very short remarks used by accountants for personal information. Implementation of content analysis using cosine similarity showed that identification of the type of operation using the comments of accountants is very likely. Further comment content analysis and financial data analysis showed that it could be expected to reduce the number of potentially suspicious operations significantly: analysis of more than half a million financial records of Dutch companies enabled the identification of 0.3% operations that may be potentially suspicious. This could make human financial auditing easier and more robust task.

**Keywords:** natural language processing; semantic similarity; cosine similarity; parsing; outliers detection

## 1. Introduction

Financial auditing is a complex and sophisticated process. A typical traditional approach is to perform financial auditing and other financial control operations by human specialists. It should be noted that high-quality financial control and auditing could be performed by highly qualified financial specialists. These specialists are expensive. An even bigger problem is that highly qualified financial specialists are scarce. This makes high-quality financial control complicated and not always possible.

One of the main difficulties in financial auditing and control is typically a huge number of operations that needs to be checked. Usually, large and even medium size companies perform large numbers of financial operations. These numbers are often large even when performed daily, while the typical audit is performed once a year or at least once in a quarter. For the human expert, it is a hard task to overview and to check all the operations simply due to the huge amount of them. The huge numbers of operations often cause errors made by accident. With a growing economy and in the process of globalization, the number of financial operations performed by companies also has the trend to grow over time.

The problem of the complexity of financial control has generated the will and the desire to develop automatic or semi-automatic tools. The range of those tools varies between fully

automatic machines using principles of artificial intelligence and auditing support tools that can be used by human auditors as a supportive element during the process of financial examination. There are hundreds if not thousands of automatic auditing and financial control tools on the market now. These products vary in their functionalities and level of sophistication, as well as the level of automation they are using. Many tools require other specialized software tools (e.g., appropriate ERP systems), which is not always available for both the company and the auditors performing checks. The fact that skilled and qualified financial auditors still are in high demand is a good indicator that automatic tools are far from perfect and far behind human capabilities in this area.

The particular problem is the identification of fraudulent or at least irregular financial operations. This is a very acute problem: according to auditing company PriceWater-houseCoopers survey "PwC's Global Economic Crime and Fraud Survey 2022", 46% of surveyed organizations worldwide reported experiencing fraud, corruption, or other economic crimes in the last 24 months [1]. This means that almost every second organization has faced fraud or other financial crimes in recent years. This number remained relatively stable in the last five years, but the executives in at least 53 countries or regions said that fraud or corruption is on the rise in their area. The big problem, in this case, is that fraudulent operations need to be identified as early as possible to minimize damage. Often identification of fraud at later stages cannot prevent significant economic losses, which in some situations could be simply huge. It means that sort of early warning systems—tools that can help indicate potential financial irregularities faster—are of great importance and demand.

Accountants often leave brief comments near records about the performed financial operation in ledger books. These comments are usually very short. There is no strictly specified way the accountant needs to comment on their record because they are for personal use only and are not legally binding. At the same time, these comments often contain useful information. This paper deals with the implementation of NLP-based models to identify potential irregularities in companies' ledger books. Section 2 presents related works. Section 3 presents proposed models to be used for accountant comment analysis with the aim of identifying irregularities in financial operations associated with the comment, as well as the possibility of identifying the operation itself. Section 4 presents the discussion about the achieved results and the prospects for future work.

## 2. Related Work

There are numerous studies devoted to finding financial data irregularities or even fraud in the financial documents of business companies. Not surprisingly, the vast majority of these studies are devoted to the purely financial aspects of this problem, e.g., creating new models that can detect financial irregularities in companies' performance, but it must be noted that in recent years more and more often, studies that try to use various computer science-based approaches (e.g., using neural networks or other elements of artificial intelligence) are found. It is necessary to emphasize that most of the authors are using only numeric data available from records made by the company (sums of money, operation Ids, value of assets, and other accounting-related information). A significant class of studies are related to the implementation of machine learning techniques aiming to train them to recognize suspicious financial activities. Several such studies are worthy of mention. An influential, widely cited study has been presented by Gray and Debreceny [2]. They investigated the application of machine learning methods to detect fraud using financial statement audits. As a result of the study, the authors proposed a taxonomy that could be used as a guideline for research in this area. The taxonomy which found and combined known fraud scheme patterns with those areas that are beneficiaries of productive applications has been developed. Kirkos et al. [3] explored the effectiveness of several techniques for data mining, to use them to classify firms' fraudulent and non-fraudulent financial statements. As a result, factors associated with fraudulent statements were identified. Neural networks, decision trees, and belief networks were used to assist

the auditor and were able to achieve relatively high accuracy. Ravisankar et al. [4] used an even wider set of machine-learning techniques for similar purposes. In their study, such techniques as multilayer neural networks, support vector machines, genetic programming, group method of data handling, logistic regression, and probabilistic neural network were used to identify companies that had committed financial statement manipulations. The methods were tested on a dataset from 202 Chinese companies. This study was essentially concentrated on features that could be used to identify fraud. Dutta et al. [5] used most of the popular machine-learning techniques to discover fraudulent financial statements. The methods used were a decision tree, artificial neural network, naïve Bayes, support vector machine, and Bayesian belief network classifiers. The study was designed in two stages. First, they mainly concentrated on finding financial fraud within a single enterprise with the help of machine learning. Then the focus was given to detecting such fraud within a whole business using diverse data. Omar and colleagues [6] developed a neural network-based mathematical model for comparing risky and fair companies selected from among small market capitalization companies in Malaysia. Ten financial ratios were used as fraud risk indicators. They were applied to predict fraudulent financial reporting using neural networks. Their study showed that the proposed neural networks-based method works better than other statistical techniques often used to predict fraudulent financial reporting: higher accuracy was achieved. The work of Hajek and Henriques [7] is important since they well demonstrated the use of ensemble-based methods and that they are superior in true positive rate (fraudulent cases are recognized as fraudulent correctly).

As was mentioned above, all these studies used only numeric financial parameters used in financial accounting to detect financial irregularities. The importance of the studies is in the fact that they showed that by using machine learning techniques, it is possible to achieve a relatively high detection rate of suspicious financial activities. The drawback of these studies is that other information than various financial indicators was left unused. In addition, some financial indicators are available only sometime later (e.g., at the end of the year or the end of the quarter), which makes it hard to identify suspicious financial activities as soon as possible.

Another group of studies is related to the application of natural language processing techniques for various financial tasks. Some of them concentrate on fraud and irregularities detection too, e.g., Goel [8] created a methodology to proactively identify means to detect fraud by investigating the qualitative content of annual reports with the help of natural language processing techniques. The proposed methodology was based on the application of SVM. The verbal content and the presentation style of the qualitative part of the annual reports were explored. The aim was to look for the linguistic features that could distinguish fraudulent reports from non-fraudulent annual reports. The main result of the study was the conclusion that to detect fraud, it is important to investigate and use the qualitative content of the report, too, since the textual content of these reports contains more information than the financial ratios only. The financial ratios can be easily falsified. Further development of the previous work has been presented by Goel et al. [9]. The results obtained showed that the application of linguistic features is an effective tool to detect fraud. It was possible to increase the prediction accuracy from baseline results of 56.75 percent obtained using an earlier proposed NLP-based method and a "bag of words" principle to 89.51 percent accuracy obtained by applying linguistically motivated features inspired by informed reasoning and when domain-related knowledge was incorporated. Wang and Wang [10] presented a hierarchical clustering approach based on linguistic features to identify fraudulent financial reports and obtained superior results (accuracy of 85–90 percent). However, the drawback of the study was that only five reports were used. Seemakurti et al. [11] described a method for the detection of fraud in financial documents using machine learning techniques to classify a collection of texts based on their topic-document frequency matrix. The authors used annual reports that give a detailed summary of a company's financial activities. Such reports are required to be presented each year. They proposed and explored a new method to recognize fraudulent and non-

fraudulent reports using the assumption that topical differences must exist between these report types. Using the LDA, they obtained the document-topic distribution matrix. The Dirichlet probability of each document belonging to each topic was derived from this matrix. The analysis of the results allowed them to conclude that the levels of accuracy had been reached that could be expected by common text classifiers. Soong and Tan [12] used similar techniques for sentiment analysis in financial data. The two-stage process has been proposed by Chen [13]. First, two decision tree algorithms—namely, the classification and regression trees (CART) and the Chi-squared automatic interaction detector (CHAID)—are used to find the most important variables for further analysis. Then a combination of CART, CHAID, Bayesian belief network, support vector machine, and artificial neural network is applied. The aim of such a combination is to construct models for fraudulent financial statements using derived variables. The results showed that the performance of the CHAID–CART model is the most effective. The overall accuracy was 87.97% in this study, while in some cases, it reached even 92.69%. An overview of natural language processing methods with applications in finance and accounting was presented by Fisher, Garnsey, and Hughes [14]. Sifa et al. [15] presented the Automated List Inspection (ALI) tool. This tool implements methods from different areas including machine learning and natural language processing while combining them with domain expert knowledge. The aim of this combination is to automate financial statement auditing. The developed tool is a content-based context-aware recommender system. The system transforms relevant text passages from the notes to the financial statement and comparing with specific law regulations. The application of deep learning for text analysis is becoming more popular. e.g., Craja, Kim, and Lessman [16] applied the combination of information containing financial ratios and managerial comments present in companies' annual reports. They employed a hierarchical attention network (HAN) to obtain the features from the Management Discussion and Analysis (MD&A) section of the previously-mentioned annual reports. The results showed that their model achieved considerable improvement in AUC compared to the benchmark models (92% vs. 85%). The main element of their research was the implementation of hierarchical attention networks (HAN) proposed by Krankel and Lee [17]. A different but interesting approach was presented by Kraus and Feurrigel [18]. They applied sentiment analysis for financial news using deep and transfer learning. This yielded significant improvements in forecasting stock price movements. All these studies used the text from annual financial reports that companies must provide at the end of the year. The long period from the moment when fraudulent operations were performed, and the report becomes available makes it hard to detect them before significant damage has been done and minimize the potential losses.

Together with such data, ledger books often have additional information, e.g., comments used by the accountants. This data almost was not used in previous studies. The reason for this is quite obvious: the comments are usually concise, and since they are not legally binding, they are prepared in an informal and often hard-to-process way. Since we believe that these comments contain much useful information, we decided to perform an investigation on which NLP methods could be exploited to identify potentially irregular operations in financial records and how many irregular operations could be found. It is hard to expect that comments can allow the identification of all irregular, suspicious financial operations. However, we expect that they can serve as a useful tool for reducing the number of records that needs to be checked by auditors and controllers.

## 3. Results

### 3.1. Problem Formulation

First, we will briefly introduce the ledger book. A ledger book is a journal in which a company maintains the data of all the financial transactions performed and other financial statements [19]. Some specific rules exist in different jurisdictions, but it could be stated that the following is universally applicable. Typically, the general ledger account is organized under the general ledger. It contains the balance sheet usually classified into multiple

accounts. Those accounts are assets, receivable and payable accounts, information about liabilities, equities, revenues, and information about taxes, expenses, profit, loss, funds, loans, bonds, stocks, salaries, wages, stockholders, and all other related financial data. All these elements are typical parameters from the point of view of financial professionals. These accounts completely describe all financial statements emerging within the company. Sometimes the ledger is called the principal book. For the financial analysis and trying to find the financial health of the company, it is necessary to relate all the information for any account available. This could be done by analyzing the ledger. Usually, a ledger account contains much additional information as well. Among the additional pieces of information are dates, particulars, and other related information, as well as comments made by accountants. The examples of the parameters contained in the ledger boom are shown in Appendix A (1). National legislation and national standards also affect the type and structure of the ledger and records contained in this book. Some countries use very strict regulations when making records while others follow less restrictive rules and more flexible ways to store the records. In addition, usually there are several types of ledgers: general ledger, sales ledger, purchase ledger, and some other ledgers too. In the modern digital world, the records from different ledger books can be easily combined within a single database and analyzed to make appropriate business decisions. It could be emphasized that in countries with stricter national regulations, it is easier to apply machine learning methods to find conclusions that could be useful for business decision-makers. However, the data from the countries that have more relaxed national regulations need to be analyzed too. In this case, some additional data processing methods could be applied and be useful too. The data used in our study was derived from The Netherlands. This country has a relatively relaxed financial accounting system which means that accountants have more freedom in the way they enter the data. Since the ledger book contains or can contain very many records (depending on a company's type and size), often accountants or other responsible people make comments. The comments are made for personal use and help to understand or clarify the financial operation described in the record. We hypothesize that these comments could be useful when analyzing the data automatically and could be used to identify some types of financial operations. The biggest problem, in this case, is the fact that comments are not standardized and any accountant is free to write the comments as they want: the comments have no legal requirements. Hence it is not necessary to use grammatically correct sentences, and the use of abbreviations is highly probable. Despite this, we hypothesize that comments have semantic meaning because otherwise, they would be useless. In this study, we will test two hypotheses. The first hypothesis is that comments could be used to recognize the type of financial operation performed. Another hypothesis that needs to be tested is that comments could help to find outliers (deliberate or unintentional) in records of financial operations fixed in the leverage book. The expectation is that similar or identical comments most often are associated with similar amounts of money. Hence it is expected that accountants will use semantically very similar or even identical comments to denote similar (maybe routine) operations and will not use identical comments next to operations dealing with completely different amounts of money (it is expected that, in this case, most often different type of operation with different comments are used). Here should be stated that this hypothesis is expected to be valid only when sums of money differ very much (e.g., hundreds in one case and hundreds of thousands in another case).

### 3.2. Methods to Evaluate the Similarity of Vocabulary

Here we will briefly present the NLP methods used in the study. The efforts have been concentrated on evaluating the possibilities of recognizing the financial operation using the accountant's comments. First, we evaluated the vocabulary of comments used in different operations. To evaluate the similarity between the vocabularies two methods were used: bag of words and cosine similarity were applied to get the general impression about proximity of different vocabularies. Cosine similarity is a simple and efficient measure to

show the similarity between different vocabularies and hence the potential to recognize the financial operation using the comments. Then tf-idf and GloVe word embedding were used to find the accuracy of the recognition of financial operations using the accountant comments only. This could serve as a fast way to identify the possible mistakes made by accountants or other types of irregularities. Additionally, we checked how the comments next to the sales and purchases operations could be used to identify possible irregular sales or purchases (typically, such operations form the largest part of all financial operations in many business companies). The main hypothesis was that semantically similar comments need to be associated with similar amounts of money.

One of the best initial ways to obtain a general understanding of how different sets of sentences or other pieces of texts differs is to count the term frequencies in those texts (bag-of-words approach). The method is fast and can provide good primary insights into the potential to use NLP methods for operation recognition using comments. At the same time, it needs to be noted that the method also is very simple and flexible. It could be used in many ways and many applications for feature extraction from textual documents. A bag-of-words is a characterization of text using the occurrences of words in a particular document or groups of documents [20,21]. It involves two things:

(1)　A vocabulary of known words.
(2)　Known words presence measure.

The name bag of words is used because information about the order of words in the document or corpora is discarded as well as the information about the structure of the document or part of the document. The model takes into account only if known words occur in the document or not but not in which place of the document. In the case of accountant comments, this fact is not so important because comments are typically short expressions, usually no longer than five lexical units (words or abbreviations). Intuition tells us that similar documents (having similar bags of words) need to have similar content too. Moving further, we can expect that from the content alone, we can obtain more or less information about the meaning of the document. At the same time, this method could be treated as simple and complex. The complexity arises from two factors: how to design the vocabulary of known words (or tokens) and rank known words [22].

Another NLP technique used in this study was cosine similarity. Here we used cosine similarity to measure the similarity between comments associated with different types of financial operations. Mathematically cosine similarity measures the proximity between two vectors in an inner product space. This similarity is expressed by calculating the cosine of the angle between two vectors. The cosine tells if two vectors are pointing in approximately the same direction. It is a well-known and widely used technique that is popular in trying to evaluate the similarity between documents. In this way, a document can be described using very many parameters, each of them describing the frequency of a particular word or phrase in the analyzed document. Each document is represented by the term-frequency vector. In mathematical terms, this could be expressed as follows [23].

$$similarity = cos(\theta) = \frac{A \times B}{||A||\,||B||} = \frac{\sum_{i=1}^{n} A_i B_i}{\sqrt{\sum_{i=1}^{n} A_i^2}\sqrt{\sum_{i=1}^{n} B_i^2}}$$

Mathematical cosine similarity means similarity between two non-zero vectors of an inner product space that measures the cosine of the angle between them. The cosine of $0°$ is 1, which is less than 1 for any angle in the interval $(0, \pi]$ radians. It is thus an evaluation of orientation but not the magnitude: two vectors with the same orientation have a cosine similarity of 1, two vectors oriented at $90°$ relative to each other have a similarity of 0, and two vectors diametrically opposed have a similarity of $-1$, independent of their magnitude.

The advantage of cosine similarity is that even if the two relatively similar documents are situated far from each other when counting the Euclidean distance (e.g., due to the size of the document), they still could be oriented closer to each other by measuring the angle [24,25]. In fact, the smaller the angle, the higher the cosine similarity measure will be.

If the two sets of expressions or two different dictionaries are identical, the cosine similarity will be equal to 1, and if they are completely different (there are no identical words in the comments left next to different types of financial operations), the cosine similarity will be equal to 0.

To recognize to which class of financial operations a comment belongs, Tf-idf measure was used. TF-IDF consists of two components, term frequency and inverse document frequency. Term frequency can be determined by counting the number of occurrences of a term in a document. IDF is calculated by dividing the total number of documents by the number of documents in the collection containing the term. It is useful for reducing the weight of terms that are common within a collection of documents. Then the tf-idf coefficient was calculated using standard formulas:

$$tf = \frac{number\ of\ times\ the\ term\ appears\ in\ document}{total\ number\ of\ terms\ in\ the\ document}$$

$$x = log\left(\frac{number\ of\ the\ documents\ in\ the\ corpus}{number\ of\ documents\ in\ the\ corpus\ contain\ the\ term}\right)$$

$$tf\text{-}idf = tf \times df$$

Tf-idf coefficient has the advantage that it is reliable and not difficult to calculate, as well as having clear interpretation.

### 3.3. Dataset Used and the Results of the Operation Type Recognition Using Comments

The dataset used in these experiments consisted of records found in ledgers of 12 Dutch companies. The dataset was obtained legally with all permissions to use data for research purposes. The initial dataset contained nearly one million records. It was found that about 38 percent of all records did not have any comments. When records without comments were removed, about 500 thousand records were left for further analysis (more precisely, 498 thousand records). Some examples of the comments found in the ledgers are presented in Appendix A (2).

First, bag-of-words have been applied to evaluate the content of the vocabularies used in different types of operations. It has been expected that rapid evaluation using BoW will allow finding if further and more detailed analysis can provide useful results. The BoW analysis led to the following observations. In many cases, it was found that with some types of operations, relatively few words are used, while some words are used very often in this type of operations. e.g., in comments with shareholders' properties, only seven different words made up 57 percent of all words used. At least one of those seven words has been used in 98 percent of records with shareholders' properties. In comments next to operations with salaries, two words were used almost always. This was the name of the month (used in 97 percent of comments from this group), and the abbreviation ljp is used exclusively in the comments next to operations with salaries. In comments next to depreciation operations, the words afschrijvingen, afschr, and afs are used in 95 percent of records. In 92 percent of comments from this group, the name of the year is also used.

On the other hand, in comments next to goods return operations, there are no dominant words. The three most often used words make only up to 8 percent of vocabulary in this case. The most frequently found word (Dutch word retour in this case) was found in only 9 percent of all comments. In a group called other operations, the digits 17, 2017, 18, 2018, 19, 2019, 20, and 2020 (these digits mean the year in this case) are found more often than in other types of operations. However, the year meaning digits make only 0.6 percent of all words found in this group of comments.

In the group called other financial operations, the words stg, molli, and payment are found in about 40 percent of comments. In these operations there are many difficult to decode (from a computational point of view) abbreviations such as t. or m. The very characteristic property of these comments is the use of words belastingdienst salari. They are found in up to 15 percent of all comments. The sales operations make up the most

significant part of all operations. The vocabulary of these comments is very diverse, but some words or abbreviations are found only next to these operations, e.g., abbreviations bv, b.v. or b.v are found exclusively in comments next to sales operations (13 percent of all comments). The abbreviation nl is found in about 20 percent of comments next to sales operations. The quite common situation is to find the address (street, city, etc.) in comments with this group of operations. This is very non-typical for other types of operations. These observations allow us to conclude that comment analysis has the potential to help identify the financial operations stored in ledger books. Cosine similarity was used to evaluate the similarity between vocabularies used in different types of financial operations. Table 1 shows cosine similarities between vocabularies used in different financial operations:

**Table 1.** The cosine similarity between vocabularies used in comments next to different types of financial operations. In this table, 1 means shareholder operations, 2—operations with salaries, 3—return operations, 4—other operations, 5—other financial operations, 6—residual operations, 7—depreciation operations, 8—sales operations.

|   | 1 | 2 | 3 | 4 | 5 | 6 | 7 | 8 |
|---|---|---|---|---|---|---|---|---|
| 1 | 1 | | | | | | | |
| 2 | 0.0 | 1 | | | | | | |
| 3 | 0.292 | 0.001 | 1 | | | | | |
| 4 | 0.058 | 0.023 | 0.038 | 1 | | | | |
| 5 | 0.018 | 0.019 | 0.057 | 0.034 | 1 | | | |
| 6 | 0.041 | 0.028 | 0.059 | 0.308 | 0.045 | 1 | | |
| 7 | 0.002 | 0.008 | 0 | 0.122 | 0.007 | 0.278 | 1 | |
| 8 | 0.007 | 0.003 | 0.104 | 0.071 | 0.155 | 0.158 | 0.178 | 1 |

The results clearly indicate that vocabularies used in comments next to different types of financial operations are essentially different. The biggest vocabulary similarities were found between residual operations and other operations, between residual and depreciation operations, and between shareholder and return operations. Even then, similarity is not very strong. In all other cases, the similarity between the vocabularies used in comments next to different types of financial operations is negligible. This indicates that the content of comments has a high potential to help recognize financial operations.

At the next stage, we performed operation recognition experiments using accountant comments. A threefold cross-validation check has been used in this experiment. We took 20 percent of randomly selected records from each of the eight types of financial operations for testing, while the remaining 80 percent of records were left for training. Abbreviations found in comments were transformed into words using the Dutch abbreviations lexicon (758 abbreviations). Abbreviations that were not present in the lexicon (32 abbreviations) were treated as an appropriate word. The recognition was performed as follows: the tf-idf value has been calculated for every word for all eight types of financial operations. If the comment had several words, the total tf-idf value of the comment was equal to the sum of tf-idf values of each word in the comment. The comment was assigned to the class of financial operations with the highest total tf-idf value. Table 2 summarizes the accuracy of recognition of financial operations using accountant comments.

The results show that comments allow for recognizing the class of performed financial operations quite accurately. This suggests that they could be used as a supportive tool when looking for potential errors made by accountants.

Even better recognition results were achieved by applying the same method for the comments within the same company. Two randomly selected companies were chosen to check the operation recognition accuracy using the comments from the same company only. An 80–20 percent cross-validation has been applied. In this case, the average operation accuracy was 99.5 percent, while the worst accuracy was 98.9 percent. Finally, we used word embedding methods to evaluate the discriminational ability of accountant comments. In this case, we applied the Global Vectors or GloVe algorithm as one of the best-suited

word embedding methods for the task [26]. The methods of experiments were very similar as in the case with the tf-idf experiments. It means that a threefold cross-validation check has been used in this experiment. Each time we took 20 percent of randomly selected records from each of the eight types of financial operations for testing, while the remaining 80 percent of records were left for training. Abbreviations found in comments were transformed into words. Then for the comments associated with each of the eight types of financial operations, GloVe training has been performed. During recognition, a comment has been assigned to the class with the highest word vector value. The results are presented in Table 3.

**Table 2.** The recognition accuracy (in %) of the financial operation type using tf-idf values. In this table, 1 means shareholder operations, 2—operations with salaries, 3—return operations, 4—other operations, 5—other financial operations, 6—residual operations, 7—depreciation operations, 8—sales operations.

|   | 1 | 2 | 3 | 4 | 5 | 6 | 7 | 8 |
|---|---|---|---|---|---|---|---|---|
| 1 | 98.6 | 0 | 0.2 | 0.3 | 0 | 0.4 | 0.5 | 0 |
| 2 | 0 | 94.2 | 3.2 | 0.2 | 1.6 | 0.6 | 0 | 0.2 |
| 3 | 0 | 0 | 99.1 | 0 | 0.2 | 0.4 | 0.3 | 0 |
| 4 | 0 | 2.4 | 0.2 | 95.7 | 0.3 | 1.1 | 0.3 | 0 |
| 5 | 0 | 0 | 2.1 | 0.4 | 91.7 | 4.5 | 0 | 1.3 |
| 6 | 0.4 | 0 | 1.2 | 3.2 | 0.8 | 93.1 | 0.2 | 1.1 |
| 7 | 0 | 0 | 0 | 0.2 | 0 | 0.2 | 99.6 | 0 |
| 8 | 0 | 0.3 | 0.1 | 0 | 0.2 | 0.3 | 0 | 99.1 |

**Table 3.** The recognition accuracy (in %) of the financial operation type using GloVe method. In this table, 1 means shareholder operations, 2—operations with salaries, 3—return operations, 4—other operations, 5—other financial operations, 6—residual operations, 7—depreciation operations, 8—sales operations.

|   | 1 | 2 | 3 | 4 | 5 | 6 | 7 | 8 |
|---|---|---|---|---|---|---|---|---|
| 1 | 99.2 | 0 | 0.2 | 0.2 | 0 | 0.2 | 0.2 | 0 |
| 2 | 0 | 96.2 | 2.4 | 0.3 | 0.3 | 0.3 | 0.2 | 0.3 |
| 3 | 0 | 0 | 100 | 0 | 0 | 0 | 0 | 0 |
| 4 | 0 | 1.1 | 0.2 | 97.1 | 0.3 | 0.8 | 0.1 | 0.4 |
| 5 | 0 | 0 | 1.8 | 0.4 | 92.7 | 4.0 | 0 | 1.1 |
| 6 | 0.4 | 0 | 1.2 | 2.1 | 0.3 | 95.4 | 0.1 | 0.5 |
| 7 | 0 | 0 | 0 | 0.1 | 0 | 0.3 | 99.6 | 0 |
| 8 | 0 | 0.3 | 0 | 0 | 0.2 | 0.2 | 0 | 99.3 |

The results obtained with GloVe need to be treated as very accurate. Still hard to say if these observations could be confirmed using data from other countries too. Table 4 summarizes the average recognition accuracies of different financial operations using accountant comments and tf-idf and GloVe methods. The GloVe accuracy outperformed the results obtained using tf-idf. It could be noted that the gains obtained using a word embedding with GloVe are significant, taking into account big differences in the vocabulary used in comments with different types of vocabulary.

**Table 4.** The average recognition accuracy of different financial operations using accountant comments and tf-idf and GloVe methods.

|   | Tf-idf | GloVe |
|---|---|---|
| Average accuracy, % | 96.39 | 97.44 |

Finding the type of operation using the comments could be useful in looking for the potential mistakes made by accountants, e.g., salary operations often are quite strictly

related to particular sums of money, and deviation from the sum could indicate a potential problem. Below we will explore how comments are related to the amount of money used in the operation. It is hypothesized that semantically similar comments most often need to be related to similar sums of money. Further, we will check this hypothesis.

### 3.4. Finding the Outliers in the Sale and Purchase Operations Using Accountant Comments

Finding irregularities is a very important element of the job of auditors and other financial specialists. Irregularities could be accidental or deliberate. Accidental irregularities may mean mistakes making the record and need to be found and corrected as soon as possible. Deliberate irregularities may indicate a significantly worse situation: sometimes, deliberate irregularities could simply mean fraud. Such behavior needs to be found and prevented before significant damage could be done to the company. Very often, finding irregularities, including fraud by human auditors and other financial specialists, is difficult simply due to the large amounts of data. Automated methods to help to find the irregularities are necessary. To find a way to identify the outliers, we formulated and checked the following hypothesis: semantically similar comments need to be associated with similar amounts of money. The sums need no to be identical, but it should be unlikely that semantically similar comments will be used in operations with very different sums of money (e.g., hundreds vs. tens of thousand). It is expected that the number of operations with semantically similar content and significantly varying sums of money associated with such operations needs to be relatively small. This could make it easier to manually check such cases and allow faster and more precise identification of cases of possible fraud.

For the analysis, we selected the sales and purchase records in the ledgers of nine Dutch companies from the previous set (three companies were removed because they had very few sales and purchase operations). The sales and purchase operations were selected because, usually, they form the largest part of operations contained in the ledger book. The total number of operations used in our experiments was close to half a million records (more than 490 thousand).

The first step was to define the ranges of the sums in the financial operations that will be treated as operations dealing with similar sums of money. This subdivision is somewhat subjective. Since the information used in the experiments was from Dutch companies and hence the operations were performed in Euros, the two different subdivisions used in the experiments are shown in Table 5. Since the boundaries between the ranges are subjective, additional checking in the areas of two neighboring ranges was performed too. We treated as semantically similar the comments that are either identical, have the same words and abbreviations coding the same words, or the same words when we removed the date or ID number present in the comment, the same words when the company name was removed, etc. Comments were processed in various ways (e.g., transformed to the lowercase only version, punctuation signs removed, etc.) to make them easier to be processed during analysis.

**Table 5.** Subdivision into the ranges: the amount within the range is treated as a similar financial operation.

| Range | Amount in 1, in Eur | Amount in 2, Eur |
|---|---|---|
| 1 | 0 | 0 |
| 2 | 0.01–9.99 | 0.01–19.99 |
| 3 | 10.00–99.99 | 20.00–49.99 |
| 4 | 100.00–499.99 | 50.00–99.99 |
| 5 | 500.00–999.99 | 100.00–499.99 |
| 6 | 1000.00–9999.99 | 500.00–1499,99 |
| 7 | 10,000–99,999 | 1500.00–4999.99 |
| 8 | $\leq 100,000$ | 5000–99,999 |
| 9 | | $\leq 10,000$ |

The analysis showed that very few records could provide important information because the operation may be a mistake or fraud. A total of 94.6 percent of all comments were unique and did not have similar enough comments to be treated as semantically the same comments or are related to very similar amounts of money. Some comments are associated with different sums from all ranges (e.g., such as comment of type, overall paid amount), and they also did not have enough potential to be treated as associated with irregular operations. Combining unique comments and comments associated with sums from all ranges, it was found that they comprise 98.4 percent of all operations. Only 1.6 percent of comments were worth more detailed analysis. Removing too general comments (e.g., the word Total), only 0.3 percent were left for detailed analysis (1248 records). These observations confirmed the expectation that automatic analysis could find and indicate those comments that may need more detailed analysis by human auditors or other financial specialists, significantly reducing the number of leverage book records to be checked. Further analysis of which comments really indicate mistakes or fraud is necessary. Since such analysis requires disclosure of data that may be a commercial secret, we were not able to do such an analysis: it could be done by financial controllers or auditors. However, automatic analysis of comments may reduce the number of operations that need to be checked manually.

## 4. Discussion

Finding the irregularities in financial documents such as ledger books of companies is a tricky and complicated task. The complexity of the task is mainly caused by the fact that ledger books and sometimes other financial documents contain very many records, which makes it very difficult to analyze them manually. Efficient checking of these records by an auditor or other financial specialist requires finding a way to reduce the number of records. Additionally, it is very important to do this as soon as possible. The majority of attempts to automate this process were concentrated on using the numeric information from the financial records (sums of money, codes of operation, etc.). Such source of information as accountant comments was essentially left untouched. The reason for this is quite obvious: being informal and usually short comments are not seen as an easy source of information. Attempts to introduce accountant comments as the source of information to identify irregular financial operations together with NLP techniques is one of the main contributions of this study. The comments have obvious advantages: they are typically introduced when making the records and hence have the potential for rapid analysis and likely for the fast finding of probable errors—deliberate or accidental.

The first group of experiments aimed at recognizing the type of financial operation associated with the comment. This could be one of the steps to identify potential errors in the ledger book. A preliminary evaluation showed that vocabularies associated with different financial operations are essentially different. This was proved by the frequency analysis of words used in comments with different operations and by evaluating the cosine similarity between the vocabularies (usually, cosine similarity was below 0.1 except in two cases where those values were 0.3 and 0.28). This suggested applying a tf-idf-based recognizer to recognize the type of financial operation using the comments. The cross-validation using recognition showed that operation could be identified with very high accuracy (for different types of operations, the accuracy was in the range of 93–98 percent). The average accuracy was 96.4 percent. Even higher accuracy has been achieved using the GloVe word embedding method. In this case, recognition accuracy varied between 93 and 100 percent, while the average accuracy was 97.4 percent. Most often, the biggest number of financial operations are sales and purchase operations. Checking the correctness of all such operations is especially tricky due to large numbers. One of the potential ways to reduce the number of them is to use comments too. The main purpose of these comments is to help the accountants understand what the operation was aimed at, and this fact needs to be exploited. The premise of the study was that similar comments would likely be associated with similar financial parameters. Many operations performed and registered in the ledgers

as sale and purchase operations could be called routine operations. This is especially true for big companies where similar sales and purchases or some other operations are performed on a regular basis. The hypothesis that similar financial operations will be associated with similar text comments was formulated and tested. The similarity in this context could mean similar amounts of sum related to the comment. Finding semantically similar comments associated with entirely different sums of money could indicate the irregularities—deliberate or intentional—in financial records. It is hard to expect to find all irregularities, but it could be expected to reduce substantially the number of records that need to be checked manually by human specialists.

Relatively simple comment processing—finding the routine operations or unique comments—reduced the number of operations that may need to be checked by more than 90 percent. Then using NLP techniques such as finding semantically similar comments and checking the sums of money associated with them allowed us to reduce the number of operations that may need manual checks to less than 0.5 percent. This, in many cases, can make manual checking feasible or at least easier available. This does not mean that all identified for additional check-up operations mean irregularities; e.g., it could be seen that some differences in sums were caused by the fact that smaller sums were part of the payment for goods or services not paid immediately and paid later as the part of the payment. At the same time, it could be seen that one expensive payment for the flowers was made while all other payments for the flowers are related to significantly smaller sums. This situation could be checked manually by the auditor or controller, and the reason for this case could be clarified.

It should be noted that the presented approach has several advantages over more common methods based on the analysis of numeric data in financial records. One of these advantages is that sometimes comments can indicate the intention of the operation performed.

The main drawback of the approach is that about 40 percent of the records (in some companies, this number reaches 60 percent) have no comments at all. This number significantly varies among the companies: different companies use comments in their own way. Some companies use more orderly comments while others are less clear and less orderly.

Compared with other studies proposed method has one advantage: it is easier to apply such a method for a fast (sometimes on-the-fly) checking of records accuracy. The majority of other irregularity identification methods are using various derived financial parameters. These parameters become available only after some period—quarterly or even annually—since they are complex and need the data from a variety of financial operations, including some less frequently performed operations (e.g., asset write-off).

The results indicate that comment processing with NLP techniques could be used as an additional source of information in AI-based financial auditing systems.

**Author Contributions:** Data curation, D.D., E.G. and M.Z.; Investigation, V.R., I.V. and K.R.; Methodology, V.R. and S.G.; Resources, A.L. and R.B.; Writing—original draft, V.R.; Writing—review & editing, V.R. and K.R. All authors have read and agreed to the published version of the manuscript.

**Funding:** This research was funded by project "Enterprise Financial Performance Data Analysis Tools Platform (AIFA)". European Regional Development Fund funds the research project according to the 2014–2020 Operational Programme for the European Union Funds' Investments under measure No. 01.2.1-LVPA-T-848 "Smart FDI". Project no.: 01.2.1-LVPA-T-848-02-0004.

**Institutional Review Board Statement:** Not applicable.

**Informed Consent Statement:** Informed consent was obtained from all subjects involved in the study.

**Conflicts of Interest:** The authors declare no conflict of interest.

## Appendix A

(1)     Data available in ledger book

     LedgerAccountCode
     LedgerAccountName

SysLedgerAccountTypeName
Rubric section
SectionCode
JournalCode
JournalName
SysJournalTypeName
FinancialYear
FinancialPeriod
EntryDate
EffectiveDate
InvoiceNumber
JournalEntryNumber
JournalEntryDescription
DebitAmount
CreditAmount
VatAmount
VatCode
VatBasePercentage
DebtorCode
CreditorCode
ImportBatchID
LedgerAccountCode
ISO JournalEntry
ID CheckNumber
JournalEntryTypeID

(2) Examples of accountant comments

Omzet Uniglas 6%
Omzet Uniglas
B.T.W. Nul tarief
B.T.W. te betalen 6%
Doorbelasting direkte inkopen 0%
Salarisdeclaratie direktievoering btw 0%
Doorbelasting poolauto's IC
Doorbelasting Accountantskosten 0%
Omzet Detachering en adviesproducten
Omzet BRIXX Licentie
Omzet VBS Licentie
Omzet BRIXX Implementatie en training
BTW te bet hoog
Omzet 6% Flora Holland
Omzet Flora Hol
Flora Holl 07062019
Inkoop Bloemen Plantion
Veilingkosten Straelen
Veilingkosten Rhein-Maas
BTW voorbelasting hoog
Debiteuren Flora Holland
Af te dragen BTW lag
Af te dragen BTW hoog 21%
Veilingkosten Plantion
Debiteuren Veiling Straelen
Omzet Overige 0%
Liquiditeitsbijdrage Flora Holland
Omzet 19% Glassinside

politie 1500 fine
Omzet Centraal Beheer
Straakvense Bloemenstee
FH Naaldwijk wk
Multi auto
Landgard Sita Plantion
Sita Plantion
Aardewerk rond dracaena
Aardewerk rond chamedorea
Telefoon
Telefoon/internet kosten 07-2017
Communicatiekosten Ctrack
Zink ovaal outdoor
Zink ovaal chamedorea
Retour Transport 30 sept
Correctie op faktuur 2019425 teveel berekend
Rhein Maas wk 37 2019
Rhein Maas wk 38 2019
Phalaneopsis wit/mix
Bijdrage bedrijfsbezoek BeBlo/ Janssen staal 9-04-2019
Huur buitenterrein Bonksel juli
Ceramic rond chamedorea
Aan u geleverde boeketten
Diversen planten
Behandeling via Qitouche
werken uitgevoerd door derden
vooruitbetaalde kosten tuinonderhoud
Containerkosten
MAXOL: Juli 2017 Salaris + Administratie
Bazaar:SaaS Service Fees Membership
UPS: UPS returns+onleverbare returns
Public transport aug/sept 2017
Telefoonkosten 09-2020
Kantoorkosten 09-2020
LinkLabs: Samsung en Philips
PERKA: Philips Garment Care brief

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
