# Peer review of "Identifying Irregular Financial Operations Using Accountant Comments and Natural Language Processing Techniques"

_applsci, doi:10.3390/app12178558_

Round 1

Reviewer 1 Report

The paper by (Rudzionis et al) tries to identify irregular financial operations using analyses of the comments left by accountant. The comments are analysed using natural language processing (NLP) technique. The main idea behind the work is to try to automise financial auditing and detect potential financial operations fraud. The ambiguity of the process is mainly because most of the financial documents contain different records and graphical schemas which makes very difficult to analyze them without properly cleaning and re-organizing the content into different types. The study focuses on only the textual comments and therefore a certain amount of important data is left without proper analyses. 

Author Response

Thank you for the valuable comments and insights, which we hope allowed us to improve the paper. Here are a brief response to the comments in the review

Comment

Response

The study focuses on only the textual comments and therefore a certain amount of important data is left without proper analyses. 

In this study, one of our main goals was to propose a method which can help to detect possible accidental errors or deliberate wrongdoing as fast as possible. The text comments are well suited for such tasks since usually they are made together with the record itself. Many other parameters appear later such as e.g. KPI coefficients. Also, the comments are rarely used in similar systems. So we decided to look for the way in which the comments could be used best for this type of task.

In the paper, we made changes to the conclusions trying to clarify this.

Reviewer 2 Report

The method requires more detail elaboration of the application. as a content analysis study, data are strongly needed to advocate the analysis

Author Response

Thank you for the valuable comments and insights, which we hope allowed us to improve the paper. Here are a brief response to the comments in the review

Comment

Response

The method requires more detailed elaboration of the application. as a content analysis study, data are strongly needed to advocate the analysis

The paper has been significantly rewritten trying to describe better the methods, experiments and data examples provided

Reviewer 3 Report

This paper presents implementation of NLP based algorithms to identify irregular financial operations using comments left by accountants. The comments are freely written and usually very short remarks used by accountants for personal information. Implementation of content analysis using cosine similarity showed that identification of the type of operation using the comments of accountants is very likely. Further comment content analysis and financial data analysis showed that it could be expected to reduce the number of potentially suspicious operations significantly: analysis of more that half a million financial records of Dutch companies allowed to identify 0,3% operations that may be potentially suspicious. This could make human financial auditing easier and more robust task.

The paper is not well-written:

1)It would be better to give a whole picture of the proposed method in section 3. It is hard to follow the proposed method in current version.

2)The data set and experimental results are not shown in a clear way.

3)The related work part is lack of organization. 

4)It would be better to give some data samples to help readers to know the data set used in this research.

The similarity calculation method is somehow simple. Suggest to involve embedding based method.

The evaluation of the proposed method is kind of weak. Line 390 mentioned that "only 0.3 percent were left for detailed analysis". It would be better to show how many real "irregularities" in the spotted 0.3 percent.

It would be better to compare the performance of the proposed method with previous works.

Minor issues:

1)Line 309 mentioned that "their information value is only about 0.6". What's the meaning of information value and how to calculate it?

2)The English writing of the draft could be improved further. There are some grammar errors. For example: line 372 "We treated as semantically similar comments that are either or identical, or has the".

Author Response

Thank you for the valuable comments and insights, which we hope allowed us to improve the paper. Here are a brief response to the comments in the review

Comment

Response

The data set and experimental results are not shown in a clear way.

The experimental results and data set have been rewritten significantly

The related work part is a lack of organization.

This part of the paper was rewritten

It would be better to give some data samples to help readers to know the data set used in this research.

The examples of the data now are provided in the annexes

The similarity calculation method is somehow simple. Suggest to involve embedding based method.

We added the results of some experiments done several weeks ago using tf-idf to recognize the financial operation using comments

The evaluation of the proposed method is kind of weak. Line 390 mentioned that "only 0.3 percent were left for detailed analysis". It would be better to show how many real "irregularities" in the spotted 0.3 percent.

This is hard for us to find out which of the suspicious operations were real irregularities. To do this we need to access the full records of commercial companies. We asked via partners (developers of software products for accountants) that some companies would check their records for potential mistakes in some operations provided by us. The request was sent to three companies but only one responded. They told that of the 44 warnings generated using the proposed method 8 cases were real mistakes made by accountants. But since this information was done by third people and we weren’t able to control the accuracy we would like to abstain from giving this info in the paper.

It would be better to compare the performance of the proposed method with previous works

Several comparisons were added

1)Line 309 mentioned that "their information value is only about 0.6". What's the meaning of information value and how to calculate it

Corrected. This was the translation error from our internal messages during the research

The English writing of the draft could be improved further. There are some grammar errors. For example: line 372 "We treated as semantically similar comments that are either or identical, or has the".

Significant English editing and rewriting has been done

Round 2

Reviewer 3 Report

1)Although the modified version used TFIDF for similarity calculation, it is still a simple method. Suggest to involve embedding based method.

2) Suggest to make the comparison results for different  similarity calculation methods clear. For example, use a table to show all the results with different  similarity calculation methods.

Author Response

Thank you for the valuable comments and insights, which we hope allowed us to improve the paper. Here are a brief response to the comments in the review

Comment

Response

Although the modified version used TFIDF for similarity calculation, it is still a simple method. Suggest to involve embedding based method.

We added word embedding. To achieve the results faster we used GloVe asnd Quanteda package. Word embedding really surpassed tf-idf and surpassed more than we expected (having in mind the already high accuracy has been achieved)

 Suggest to make the comparison results for different  similarity calculation methods clear. For example, use a table to show all the results with different  similarity calculation methods.

New table for the comparison was added

Round 3

Reviewer 3 Report

The authors have modified the paper according to the comments from reviewers. I have no further comments.

Author Response

A series of changes were introduced to various parts of paper trying to make easier to read and more clear